# Variation in wood density across South American tropical forests

Wood density is a critical control on tree biomass, so poor understanding of its spatial variation can lead to large and systematic errors in forest biomass estimates and carbon maps. The need to understand how and why wood density varies is especially critical in tropical America where forests have exceptional species diversity and spatial turnover in composition. As tree identity and forest composition are challenging to estimate remotely, ground surveys are essential to know the wood density of trees, whether measured directly or inferred from their identity. Here, we assemble an extensive dataset of variation in wood density across the most forested and tree-diverse continent, examine how it relates to spatial and environmental variables, and use these relationships to predict spatial variation in wood density over tropical and sub-tropical South America. Our analysis refines previously identified east-west Amazon gradients in wood density, improves them by revealing fine-scale variation, and extends predictions into Andean, dry, and Atlantic forests. The results halve biomass prediction errors compared to a naïve scenario with no knowledge of spatial variation in wood density. Our findings will help improve remote sensing-based estimates of aboveground biomass carbon stocks across tropical South America.

Understanding spatial and temporal variation in forest biomass carbon stocks is critical for numerous applications and research questions, including national carbon stock inventories [e.g. ref. 1], assessments of forest responses and recovery from disturbance[2–4], and investigation of climate feedbacks [e.g. ref. 5]. However, quantifying the distribution of aboveground live carbon stocks across the tropical forest biome remains challenging. Despite decades of fieldwork[6] and investment in satellite and airborne remote sensing to measure canopy structure with Lidar or vegetation volume through radar scattering[1,7], there is still considerable uncertainty about the amount and distribution of aboveground carbon in tropical forests. Indeed, marked differences among recent global maps of biomass carbon[8–10] reflect the challenge of large-scale calibration and validation across tropical forests.

The challenge partly arises because remote-sensing approaches, which allow large-scale and spatially continuous measurements, cannot provide all the information available from ground-based surveys. Wood density is a fundamental determinant of tree biomass[11–13], and estimating it requires skilled botanical surveys to identify trees.

Airborne and satellite remote-sensing methods provide measurements that allow estimates of tree height or volume, but not their identity or wood density[14]. While some inferences about taxonomic composition can be made from hyperspectral imagery [e.g. refs. 15, 16], this remains limited compared to what can be obtained by a ground-based botanical survey. Lack of wood density information can lead to marked discrepancies between remote sensed and ground-based estimates of aboveground biomass[17], including spatial biases in aboveground biomass estimates of around 30% even within a single country[18].

Future improvements in remote-sensing-based forest biomass maps therefore require improved knowledge of spatial variation in tree wood density. The need to tackle this huge challenge is especially important in South America. Not only are tropical rain forests here the most extensive in the world, but they also include many of the most productive and carbon-rich forests on Earth[19,20] and large carbon sinks and fluxes [e.g. 21–23]. The nature of the challenge is also most profound in South America, as ~40% of Earth's 73,000 tree species are found in forests here[24]. Amazonia alone is home to at least 15,000[25], and beyond

✉e-mail: martin.sullivan@mmu.ac.uk

the Amazon marked differences in species composition pertain across South America's diverse forest ecosystems[26–28].

While the proximate driver of spatial variation in wood density is turnover in species composition, it may ultimately relate to environmental gradients, as wood density is an important ecological trait mediating species responses to the environment. High wood density trees experience lower mortality risks[29,30], but as dense wood is costly to produce there is a trade-off between producing less dense wood and growing faster, and producing denser wood and having lower mortality risk[31]. This lower mortality risk may arise through resistance to mechanical breakage[32], the dominant cause of tree death in Southern and Western Amazonia[30,33], although resistance to breakage may be offset by greater flexibility of low wood density species[34]. Wood density is also linked to drought sensitivity, as higher wood density predicts lower vulnerability to cavitation[35,36] and resilience of growth to drought[37], although these relationships are influenced by the allocation of xylem space to different tissues[38]. Species with high wood density are therefore likely to tend to be more tolerant of environmental stresses such as drought, while the growth advantage of low wood density species may be most marked in competitive environments (e.g. with high soil fertility) and in frequently-disturbed forests where rapid colonisation of gaps is key (e.g. unstable soils). There is some empirical evidence to support these theoretical predictions. For example, Chave et al.[32] found that wood density varied across North and South America with gradients of temperature and precipitation, while Quesada et al.[39] found that wood density was lower on more fertile and more poorly structured soils, as well as tending to be higher where precipitation was lower and temperatures were higher (i.e. greater potential for drought stress). These studies highlight the potential for improved prediction of spatial variation in wood density by incorporating relationships with environmental variables. However, it is unknown how such different drivers acting at multiple spatial scales combine to influence variation in wood density across tropical South American forests.

We leverage our extensive collective effort to measure and identify trees in forests across South America to describe spatial variation in community wood density, and use relationships with environmental variables to map estimated wood density at 1km resolution across tropical and sub-tropical South American forests. This builds on early descriptions of spatial variation in wood density [e.g. [17,40]] by utilising newly available forest plot data, and expands the analysis to include non-Amazonian forests, providing a resource to support remote-sensing analyses quantifying aboveground biomass.

## Results

### Variation in wood density

Basal-area weighted wood density varied two-fold across tropical and sub-tropical South American forests (mean = 0.63 g cm$^{-3}$, Fig. 1). Wood density varied significantly between regions (linear model, $F_{7,973}$ = 71.6, $P < 0.001$, $R^2$ = 0.34). Within Amazonia, forests in East-Central Amazon and the Guiana Shield had the highest wood density on average, followed by the Brazilian Shield, with the lowest wood density in western areas (Fig. 1b). Wood density in the Atlantic Forest was similar to that in the Brazilian Shield. Dry forests tended to have high wood density, but there was a cluster of plots (distributed across dry forest areas) with some of the lowest wood density in the dataset. Montane forests had the lowest average wood density (Fig. 1b).

### Spatial patterns in wood density

To understand this variation, models (generalised additive models [GAMs] and random forests) were constructed with three sets of explanatory variables; (1) latitude and longitude only, (2) environmental (climate, soil and topography, see Table 1) variables only, and (3) both environmental and spatial variables. Longitude was the most important explanatory variable across all models it was included in (Fig. 2), with a west-to-east gradient in increasing wood density (Fig. S1). Latitude, soil texture and soil chemistry (cation exchange capacity and pH) were the next most important variables (Fig. 2), although there were differences between models, with greater importance of pH and soil texture in GAM models compared to random forests (Fig. 2). Wood density decreased with cation exchange capacity in the GAM and random forest models without spatial covariates (Fig S1), but these relationships were weaker when latitude and longitude were included (Fig. 2, Fig. S1). While climate variables were generally less important than soil variables, their importance varied between models. Mean annual temperature was the most important climate variable when spatial covariates were not included, while cloud frequency and maximum cumulative water deficit were more important climate variables in the GAM with spatial covariates (Fig. 2). GAMs modelled positive relationships with wind speed and negative

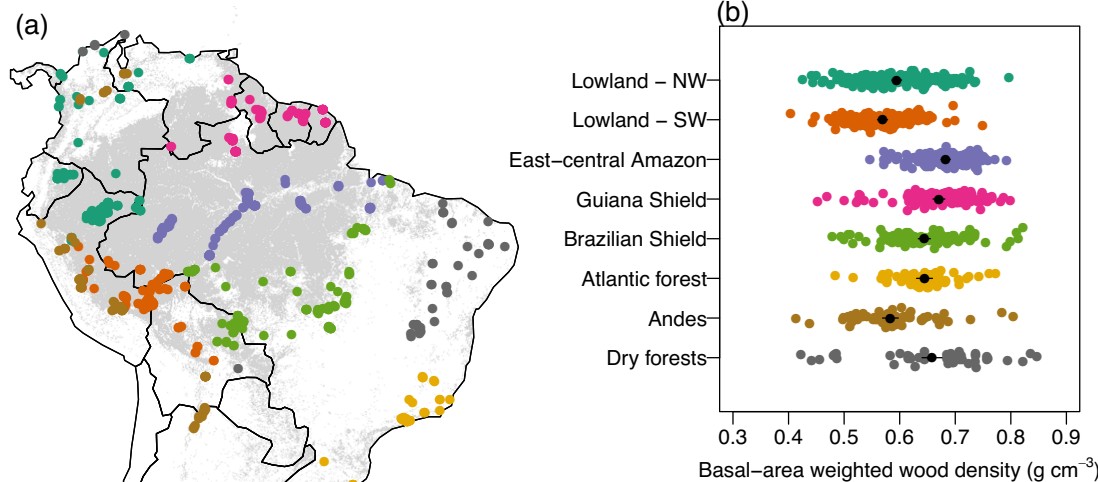

**Fig. 1 | Variation in wood density across South American tropical forests.**
**a** Location of forest inventory plots where wood density was quantified. **b** Variation in basal-area weighted wood density between regions. Colours in **a** relate to regions in (**b**). N = 981 plots (Lowland-NW = 182 plots, Lowland-SW = 168, East-central Amazon = 205, Guiana Shield = 123, Brazilian Shield = 119, Atlantic forest = 69, Andes = 71, Dry forests = 44). Black points show mean values in each region estimated by a linear model with wood density as the response variable and region as the explanatory variable; lines show 95% confidence intervals from that model.

**Table 1 | Explanatory variables used in this study and their hypothesised link to wood density**

| Variable | Potential effect on wood density | Source |
|---|---|---|
| Mean annual temperature (bio1) | Greater potential for drought stress at higher temperatures, so community-weighted mean wood density expected to increase with temperatures. | Worldclim V2[59] |
| Mean annual precipitation (bio12) | Greater drought stress (and hence expectation for higher wood density) when precipitation lower. | Worldclim V2[59] |
| Maximum cumulative water deficit (MCWD) | Higher wood density expected when moisture availability most limited. | Calculated using data from Worldclim V2[59] and TerraClimate.[60] |
| Mean wind speed in the windiest month | Proxy for potential for wind damage. High wind speeds could favour high wood-density species. Alternatively, frequent disturbances[77] could favour low wood density pioneers. | Worldclim V2[63] |
| Cloud frequency | Frequent cloud could reduce drought stress from evapotranspiration. | 61 |
| Lightning frequency | Potential to cause canopy gaps favouring pioneers.[77] Trunk properties influence impacts of lightning strikes.[78] | TRIMM LIS Very High Resolution Gridded Climatology.[79] |
| Soil cation exchange capacity | A proxy for soil fertility (other metrics such as total exchangeable bases would be preferable but are not available in gridded form across the study area). More fertile soils are expected to favour faster life-history strategies, leading to lower average wood density. | SoilGrids[62] |
| Soil pH | Potential control on tree species distributions, with specialist communities in low or high pH environments. | SoilGrids[62] |
| Depth to rock | Shallower soils are potentially unstable, and frequent disturbances would be expected to favour pioneer species, which tend to have low wood density. | SoilGrids[62] |
| Soil texture | Relates to soil stability (more disturbances leads to more low wood density pioneers) and to soil moisture holding capacity. | SoilGrids[62] |
| Topography - rugosity | More disturbances likely on steeper slopes, which favours pioneers, which tend to have low wood density. | Calculated from SRTM V4[64] |
| Topography - slope | More disturbances likely on steeper slopes, which favours pioneers, which tend to have low wood density. | Calculated from SRTM V4[64] |
| Topography - HAND | Height above nearest drainage relates to soil drainage. Wood density may be lower in soils that retain moisture due to lower drought stress. | Donchyts et al.[65] |
| Spatial coordinates | Wood density is expected to vary independently of the environment due to biogeographic variation in species distributions. Additionally, spatial coordinates can capture variation in wood density caused by environmental factors not included in the analysis. | |

relationships with lightning frequency, but these relationships were less evident in random forest models (Fig. S1). Topography, height above nearest drainage and rock depth had limited effects in all models (Fig. 2).

The west-to-east gradient in Amazonia of increasing wood density was evident in predictions of spatial patterns in wood density from all models, with the highest predicted values along the East and Central Amazon basin and in the Guiana Shield (Fig. 3). Some differences were evident between models (Fig.S2, S3), for example the GAM trained on environmental explanatory variables predicted a region of high wood density in the south-east of Brazil's Amazonas state (Fig. S2). All models predicted high wood density in seasonally dry tropical forests to the south and east of the Amazon Basin, but lower values were predicted in northern South America. As well as capturing broad-scale gradients, models predicted substantial local-scale variation in wood density, which are likely to reflect local variation in soil characteristics (Fig S4). However, when comparing observed and predicted wood density values in plots, it was notable that models predicted a more restricted range of wood density values (Fig S5). Uncertainty in predictions between models varied spatially, with greater uncertainty in Andean montane forests, southern Venezuela, and the south-east fringes of the Amazon basin, and strong agreement between models in part of Western Amazonia and the Guiana Shield (Fig. S3).

### Performance of models

When tested using cross-validation, predicted values of wood density were positively correlated with observed values for all modelling methods (r = 0.62 – 0.75, coefficient of determination = 0.37–0.57 Table 2), with mean prediction errors (i.e. the difference between observed and predicted stand-level wood density values) of 0.049-

0.057 g cm$^{-3}$ (Table 2). These prediction errors were lower than would be obtained by comparing the overall mean wood density in our dataset with observed values (prediction error = 0.105 g cm$^{-3}$). When model predictions were tested on independent spatial clusters (e.g. fitting models without dry forests, and testing predictions on dry forests), correlations with observed values were lower but remained positive on average (r = 0.272 for ensemble, Table 2), and prediction errors were larger (0.068 g cm$^{3}$ for ensemble, Table 2) but on average were still lower than if an overall mean had been used (Table 2). However, negative coefficients of determination for models tested on independent clusters (Table 2) indicates that differences between model predictions and observed values were larger than if the mean for that region was used.

Using the database mean value for wood density to estimate plot-level carbon stocks led to a median error of 8.4% (interquartile range = 4.0 – 14.0%), while using the observed plot-level mean wood density value resulted in a median error of 0.8% (interquartile range = 0.4 – 1.7%). Using the ensemble mean of model predictions resulted in a median error of 4.5% (interquartile range = 2.2-8.7%), with individual models having median errors between 4.5% and 5.4%. Our model predictions therefore lie close to midway between the naïve scenario with no knowledge of spatial variation in wood density and the best-case scenario with perfect locally-based knowledge of spatial variation in wood density.

### Discussion

We assembled an unprecedented dataset of variation in wood density within and across the biomes of Earth's most forested and tree-diverse continent, and employ multiple methods to relate wood density to environmental and spatial variables to produce predictions of wood

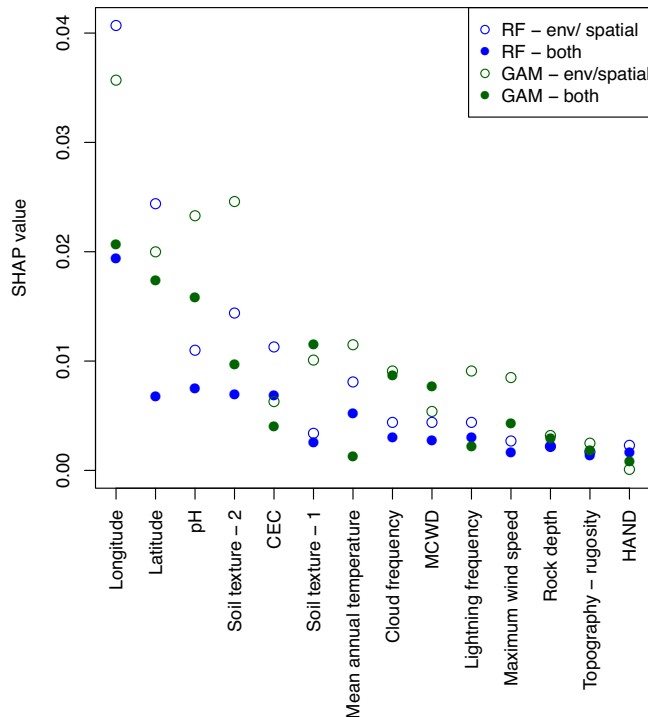

Fig. 2 | **Contribution of explanatory variables to models of spatial variation in wood density.** Global Shapley additive explanations (SHAP values) have been calculated for random forest (RF, blue) and generalised additive models (GAM, green) fitted with either just environmental or spatial variables (env/spatial, open circles) or to both environmental and spatial variables (both, filled circles). Higher SHAP values indicate a greater contribution of a feature to model predictions. Variables are ordered on the x-axis based on their mean SHAP value across models.

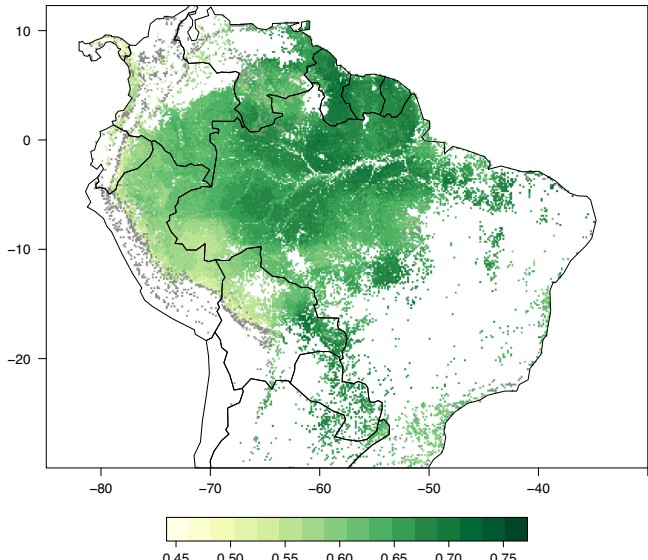

Fig. 3 | **Predicted spatial variation in basal-area weighted wood density across tropical and sub-tropical South America at 1km resolution.** Ensemble predictions averaging the six input models are shown. Predictions for individual models are presented in Fig. S2, with inter-model uncertainty in Fig. S3. Forested areas outside the area of applicability of model predictions are shown in grey – see Fig. S2 for model predictions without this exclusion, and Fig. S7 for alternative definitions of areas of model applicability.

density and associate error estimates in mature forests across tropical and sub-tropical South America. These provide important new products for the remote sensing community, approximately halving errors in carbon stock estimates compared to having a single mean value of wood density. Additionally, our work advances the understanding of spatial variation in wood density by (1) mapping fine-scale variation, which field data shows is substantial[11,18] but which is not captured in previous analyses using spatial interpolation of plot data[17,40], (2) extending predictions of wood density across Andean, seasonally dry and Atlantic forests, and (3) establishing that previously described gradients in wood density across Amazonia [e.g. [41]] largely hold in this substantially larger dataset.

## Patterns and drivers of spatial variation in wood density

Previous studies have described a gradient in wood density across Amazonia, with the highest wood density in forests in the Guiana Shield, and lower values in western Amazonia[17,40,41]. While our results largely support this pattern we find that wood density in parts of Eastern and Central Amazonia are similar or higher to the Guiana Shield. Our results are also consistent with previous studies reporting lower wood density in montane forests[18,42], and indicate that while seasonally dry tropical forests on average have high wood density, dry forests can have amongst the highest or very lowest wood densities of all South American forests.

Despite the generally higher observed and predicted wood density in seasonally dry tropical forests, we did not find strong relationships between dry season water availability, temperature or cloud cover and wood density, so do not find clear support for the hypothesised effects of water limitation leading to higher wood density. This contrasts to previous studies which found negative relationships

between wood density and precipitation within Amazonia[39] and amongst tree species across the Americas[32]. It is possible that some effects of these climate variables have been attributed to latitude and longitude in our models, but this does not explain the lack of clear relationships in models without spatial variables. Furthermore, by simply considering observed wood density values, it is evident that species in forests experiencing substantial water limitation can have very high or very low wood density (Fig. S1), possibly reflecting the diversity of strategies trees have for coping with water limitation, which can result in high wood density (high vessel density) or low wood density (water storage in tissues)[43]. Wood density tends to decrease with succession in dry forests, with high wood density species facilitating the establishment of other species[44]. While all our plots were in mature forest, plots are subject to natural disturbance events which is expected to lead to substantial variation in wood density.

There was some support for wood density being correlated with factors associated with more frequent disturbances. Soil texture was identified by models (especially GAMs) as important for predicting wood density, which is consistent with the mechanism proposed by Quesada et al.[39] whereby less stable soils lead to more frequent disturbances which promote low wood density species. We note that the measures of soil properties available in gridded datasets are imperfect, and that evaluation using in situ soil data would add stronger support for this hypothesis, as well as for the relationships with soil chemistry variables (CEC and pH) identified here. Wood density also tended to be lower where lightning was more frequent, which would be consistent with lightning disturbances promoting forest turnover, but more observations are needed from forests experiencing frequent lightning, especially as this negative effect of lightning was sensitive to sub-sampling data (Fig. S6).

## Performance and applicability of model predictions

Despite the size of our dataset, it still represents a small fraction of the forested extent of tropical South America. We assessed the area we could validly make predictions to in three ways. Firstly, we identified areas where model predictions were especially sensitive to

**Table 2 | Performance of models when applied to data not used in model fitting**

| Model | Cross-validation (interpolation) | | | Spatial cross-validation (extrapolation) | | |
|---|---|---|---|---|---|---|
| | RMSE | Correlation | Coefficient of determination | RMSE | Correlation | Coefficient of determination |
| Dataset | 0.105 | 0.011 | −1.091 | 0.105 | 0.061 | −1.574 |
| RF - spatial | 0.051 | 0.749 | 0.536 | 0.069 | 0.055 | −0.192 |
| RF – environment | 0.051 | 0.736 | 0.539 | 0.077 | 0.132 | −0.449 |
| RF – both | 0.049 | 0.755 | 0.567 | 0.070 | 0.153 | −0.253 |
| GAM – spatial | 0.054 | 0.701 | 0.480 | 0.081 | 0.149 | −0.809 |
| GAM – environment | 0.057 | 0.618 | 0.369 | 0.070 | 0.212 | −0.515 |
| GAM – both | 0.053 | 0.704 | 0.474 | 0.071 | 0.219 | −0.388 |
| Ensemble | 0.049 | 0.759 | 0.567 | 0.068 | 0.272 | −0.132 |

Model performance has been assessed using k-fold cross-validation (presumed to reflect interpolation performance) and spatial cross-validation (where an entire region was removed for model testing, presumed to reflect extrapolation performance).

Model performance has been assessed as (1) root mean square error (RMSE), which indicates the average prediction error (g cm$^{-3}$), (2) the correlation coefficient between observed and predicted values and (3) the coefficient of determination [1-(residual sum of squares/ total sum of squares)]. Negative coefficient of determination values indicate that the difference between model predictions and the testing data are greater than the difference between the testing data mean and the testing data. Median values across cross-validation folds are presented.

subsampling data. Secondly, we identified areas with explanatory variable values outside the range seen in our data. Thirdly, we calculated a multivariate dissimilarity index describing environmental and spatial variables, and calculated the area of applicability[45] of our models based on the dissimilarity index values of our different training and testing datasets (Fig. S7). These mostly indicate that our models are applicable to the majority of tropical South American forests, but should be more cautiously applied to higher-elevation Andean forests. However, some dry forest areas and parts of lowland Amazonia had dissimilarity indices higher than typically observed between non-spatial cross-validation folds, meaning that non-spatial cross-validation metrics should be seen as an upper bound rather than central estimate of model performance in these areas (Fig. S7).

Model performance was substantially lower when tested using spatial cross-validation (i.e. leaving a region out of model training, and using that for model testing). Indeed, while correlation coefficients remained positive, negative coefficient of variation values indicate that models were systematically over or underpredicting wood density, leading to greater prediction errors than if the true regional mean was used (although there were still substantial improvements over using the dataset mean). This spatial cross-validation is expected to represent a lower bound of model performance[46] as it requires predictions to be extrapolated beyond the range of training data, whereas 97% of tropical South American forests had dissimilarity index values of one or less (indicating that the dissimilarity to the most similar training data point is less than or equal to the mean dissimilarity of points within the training dataset).

While models substantially improved predictions of wood density compared to just using a mean value across the dataset, improvements in models with environmental explanatory variables compared to those with just spatial explanatory variables was limited. Spatially structured explanatory variables can show good predictive skill despite having no causal effect[47], and both GAMs and random forests were capable of fitting complex relationships between wood density and latitude and longitude. When environmental variables were also included, the complexity of relationships with spatial variables reduced (Fig. S1). This indicates that spatial-only models captured environmental variation that was taken up by environmental variables when both types were included in models. In models with both environmental and spatial variables, latitude and longitude could still capture gradients caused by unmodelled environmental variables (e.g. soil phosphorus, which was not available as a fine-scale gridded variable), or capture gradients due to biogeographic history. In the former case, relationships may not extrapolate as spatial coordinates may not prove a reliable proxy beyond the range of the training data, while in the latter case spatial coordinates are more closely tied to a causal

mechanism. The relative superiority of models with both spatial and environmental variables compared to those with spatial variables alone was greater when evaluating models on spatially distinct training data (Table 2), which would be consistent with environmental variables better capturing causal mechanisms.

Previous studies have modelled relationships between environmental variables and wood density at larger (all Americas[32],) or smaller (Amazonia alone[39],) scales than this study. Larger spatial extents, and hence larger environmental gradients, means it is more likely that environmental response curves are fully characterised[48] but increases the chances of spatial nonstationary and therefore missing regional relationships[49]. We explored this by training models separately for each spatial region, and comparing predictions to models trained to the whole dataset. Predictions of regionally and dataset-wide trained models were similar (Fig. S5), which indicates that our models were sufficiently flexible to capture regional patterns.

We used gridded climate and soil data, which would have much greater measurement error than in situ values. This is expected to lead to regression dilution[48,49], where relationships with climate and soil variables are weaker than they would be if measured in situ. Comparisons of relationships with soil variables with previous studies using the smaller number of plots with in situ soil measurements[39] should therefore be made cautiously. It is also important to note that analyses relate to wood density treating wood density as a species-level attribute. However, wood density also varies within species along environmental gradients[50,51] and with stand characteristics[52]. In situ measurements of wood density are sparse, so treating it as a species-level variable was the only feasible approach for a study of this scale, but patterns could be further refined by consideration of intra-specific variation.

It is important to note that our predictions are for mature, closed canopy forests, so should not be used for secondary forests. Wood density is expected to be lower in secondary forests[2], although in some seasonally dry and montane forests wood density can decline with succession[44,53]. These differences in trajectories of wood density between forest types may be explained by differing successional mechanisms[44], so we may expect large-scale spatial patterns in secondary forests to differ from the old-growth forest patterns described here.

We provide ensemble averaged predictions (Fig. 3) alongside predictions of individual models (Fig. S2), inter-model uncertainty (Fig. S3) and their spatial applicability (Fig. S7). Using these estimates of spatial variation in wood density is anticipated to approximately halve errors in carbon stock estimates compared to a naïve scenario where only the mean wood density is known. While there is potential to improve models further to reduce prediction errors, some errors will

remain even with perfect knowledge of spatial variation in wood density, as it would still not be known which trees within a plot have higher or lower wood density. These remaining errors represent data that can only be obtained with ground-based field surveys to identify and measure every tree in a plot. In all, while our analysis reveals some of the challenges of high-fidelity biomass mapping in species rich forests it substantially advances the spatial extent, granularity and environmental range of tropical American forest wood density measurement and prediction. Our findings and maps will contribute to better remote sensing-based estimates of biomass carbon stocks across tropical South America.

## Methods

### Plot selection and field sampling

We identified and measured trees in forest inventory plots in tropical and sub-tropical South America. These plots were established and maintained by networks of researchers (RAINFOR, DBTV, COL-TREE, TROBIT, DRYFLOR, ATDN, PPBIO, FATE, RAS, MonANPeru, Nordeste, sANDES, SECO, BDFFP) using shared protocols[54], and are curated and stored in the online ForestPlots.net database[6,55]. These networks and ForestPlots.net aim to promote equitable research collaborations in tropical ecology, and the development of this study followed the ForestPlots code of conduct (https://forestplots.net/en/join-forestplots/code-of-conduct). Plots for this study were selected based on being in mature, structurally intact and closed canopy forests (i.e. excluding secondary forests, forests with a known history of logging or burning, and savannah formations). While no restrictions in terms of soil type, edaphic factors or elevation were applied, plots were filtered to exclude those in which fewer than 80% of stems were identified to genus level, giving a dataset of 981 plots (Fig. 1). For multi-census plots, we use data for the first census for comparability with single-census plots and because a higher proportion of stems were identified to species in this census in >80% of instances. Plots were predominantly established following standardised RAINFOR protocols[54] although plots varied in area (0.04 to 25 ha, mean area = 0.76 ha). In each, the diameter of all stems ≥10cm were measured at breast height (1.3m) or above buttresses or other deformities. Stems were identified by botanists to species level where possible (85.1% of stems identified to species and 95.4% to genus).

### Wood density metrics

The wood density of each stem in our dataset was estimated by cross-referencing the taxonomic identify of each tree with a database of wood density values[56]. We note that this approach does not capture intra-specific variation in wood density[57], and that even mean wood density is missing for many species[56]. Stems were matched to the mean species-level value where possible (46.4% of stems), followed by genus-level (38.6%), family-level (11.0%) and plot-level mean values (4.0%), with taxonomic matches performed using the getWoodDensity function in the BIOMASS R package[58].

Stand-level wood density can be summarised from these tree-level values in a variety of different ways, each requiring increasing amount of information about the composition of the stand. Firstly, wood density can be calculated as the mean value across all taxa present in a stand ($WD_1$). For this, we took the arithmetic mean of wood density for each taxonomic entity (i.e. the set of fully identified species, genera with indeterminate species identifications, families with indeterminate genus, and fully unidentified stems, with taxon-level wood density obtained as described above). This discounts information about taxon abundance, simply considering which taxa are present. Secondly, wood density can be calculated as the abundance weighted mean of taxa present in the stand (i.e. the mean wood density of all stems in the stand, $WD_2$). For this, we took the arithmetic mean of wood density across all stems in the plot. This includes information about abundance, but discounts information about stem size. Thirdly, the basal-

area weighted mean wood density can be calculated, which gives more weight to stems that account for a larger proportion of stand basal area ($WD_3$). For this, we took the basal-area weighted mean of wood density of all stems in the plot. This therefore includes information about the size of stems as well as their abundance. We calculated all three metrics but only present results for $WD_3$ (basal-area weighted wood density). This incorporates the most information about stand composition and is the most directly linked to aboveground biomass; all three metrics were strongly correlated (Fig. S8).

### Environmental variables

We obtained climate and soil variables that were hypothesised to relate to spatial variation in wood density (Table 1) at 1km resolution. Mean annual temperature and total annual precipitation were obtained from Worldclim version 2[59]. To represent seasonal drought stress, we calculated maximum cumulative water deficit (MCWD) as the cumulative balance between monthly precipitation (from Worldclim[59]) and potential evapotranspiration (from TerraClimate[60],). For each plot, we calculated the balance between precipitation and potential evapotranspiration in the wettest month, and then calculated the water balance in subsequent months as the difference between precipitation and potential evapotranspiration in that month plus the cumulative water balance, if negative, in the preceding month. The minimum value of this metric across the year, representing the greatest cumulative water deficit, was obtained for each plot. Cloud variables were obtained from Wilson and Jetz[61], and represent the proportion of Modis passes at each location where cloud was present. Soil variables were obtained from the SoilGrids database[62]. In situ soil data would be preferable for quantifying relationships between wood density and soil variables [e.g.[39]], but could not be used because of the need for a dataset with complete spatial coverage for extrapolating wood density values. We used soil cation exchange capacity (CEC) as a measure of soil fertility and extracted soil pH, depth to rock horizon, and the percentage of sand, silt and clay. The latter three variables were simplified into two variables as *Texture1* = ln(*Sand/Clay*) and *Texture2* = ln(*Silt/Clay*). CEC was chosen as a measure of soil fertility as it is available across the study area, but we note that it is not a perfect proxy as it includes saturation with H and Al. We included an interaction between CEC and pH to account for this (see data analysis), and also explored the sensitivity of our results to using an Amazon-only soil base cation concentration dataset[63]; predicted wood density using CEC and soil base cation concentration were strongly correlated (r = 0.97-0.99 across models). Topography was quantified from the hole-filled 90m resolution SRTM[64] as mean slope in a 200m diameter buffer around plot locations and rugosity as the standard deviation of elevations in this buffer. Height above nearest drainage (HAND) was obtained from[65]. Topography metrics were processed in Google Earth Engine[66], other metrics were processed in Rv4.2.2[67].

### Data analysis

Statistical analysis was motivated by the goal of prediction[68]. We constructed models with three sets of explanatory variables. Firstly, wood density was modelled as a function of just longitude and latitude, providing a spatial interpolation of the data. Secondly, wood density was modelled as a function of environmental variables alone. These variables have potential causal effects on wood density, but may also capture variation due to unmodelled spatial gradients. Finally, we modelled wood density as a function of both environmental variables and spatial coordinates. This latter approach potentially allows spatial gradients not included in the environmental explanatory variable set to be captured by the spatial variables, and was chosen over methods that account for non-independence of model residuals, which may be preferable if our goal was inference, as our approach allows spatial effects to be included in model predictions.

We related wood density to these explanatory variables using random forests and generalised additive models, with both approaches chosen as they can capture complex non-linear relationships between response and explanatory variables. We checked for collinearity between explanatory variables prior to training models, and found strong corelations (|r| >0.7) between slope and rugosity, slope and HAND, and between mean annual precipitation and MCWD, and therefore discarded one of each pair (slope and mean annual precipitation) from subsequent analysis. Following removal of these variables, variance inflation factors for environmental variables were all less than four.

Random forests were constructed using the randomForest R package[69]. Hyperparameters for the number of trees to construct and the number of variables to sample at each split were selected by trying each combination of hyperparameter pairs (2-8 variables to try, and 100 to 1000 trees in increments of 100), and selecting the combination with the lowest mean square error (800 trees, three variables tried at each split).

Generalised additive models (GAMs) were constructed using the mgcv R package[70]. The complexity of non-linear relationships in GAMs was selected using restricted maximum likelihood. The basis dimension, which sets the maximum complexity of smooth terms, was set to nine for environmental variables. While this allows more complex non-linear relationships than might be theoretically expected, it ensures the function space in the realm of ecologically expected relationships (e.g, unimodal) is not overly constrained. Latitude and longitude were modelled as a single interacting smooth term with a basis dimension of 50. GAMs were also fitted with a penalty term that selects variables out of the model[71], but these performed worse at predicting wood density than models without the penalty term, so were not used further. Residual spatial autocorrelation was not evident in any of the models (Fig. S9). Variable importance was assessed using approximate Shapley additive explanations values, which provide additive contributions of each feature to each observation[72]. Values were calculated using the fastshap R package[73] and summarised across the dataset to give the global importance of each variable.

We assessed model performance using both spatial and non-spatial cross-validation. These are expected to provide lower and upper bounds of predictive performance respectively[46]. For non-spatial cross-validation, data were divided into ten approximately equal sized sets, with each set left out of model training in turn and used as independent test data. A problem with this validation method is that calibration and validation data, while different, may not be truly independent because of spatial autocorrelation, so model predictive performance may be overestimated. We therefore also applied a more stringent validation procedure where data were split into spatial clusters, and one cluster left out in turn from model fitting to be used for validation. We assigned plots to one of six biogeographic regions (adapted from[67,68]). These were North-west lowland forests, South-west lowland forests, East-central Amazon, Guiana Shield, Brazilian Shield, with remaining plots (Atlantic Forest, montane forests > 1200 m asl, seasonally dry forests with < 1000 mm precipitation per year) grouped together into a sixth region. This method ensures that test data are truly independent of training data but is likely to be overly harsh as it truncates environmental gradients meaning that models are forced to extrapolate into novel environmental space. Model performance was assessed using three metrics; the square-root of the mean-square error between predicted and observed values (RMSE), the correlation coefficient between predicted and observed values, and the coefficient of determination (1-(residual sum of squares/ total sum of squares)). These were calculated for each model, and compared to a null scenario where predicted wood density values were randomly drawn from the distribution of observed values.

To assess the consequences of imperfect wood density estimates for estimates of carbon stocks, we estimated aboveground carbon stocks in each plot using (1) taxonomically matched wood density of each stem (i.e. the data available from field surveys), (2) the observed mean wood density for the plot (i.e. the data that would be available if spatial variation in stand wood density could be estimated perfectly), (3) predicted mean wood density from the different models, and (4) the dataset mean wood density (i.e. a naïve position with no knowledge of spatial variation in wood density). The aboveground biomass of each stem was estimated using the Chave et al.[13] allometric equation applied to measured diameters, the aforementioned wood density values, with height estimated based on relationships between environmental stress and height-diameter allometries[13]. Calculations were conducted using the BIOMASS R package[58]. Aboveground biomass estimates were converted to carbon using a carbon fraction of 0.456[74].

## Mapping wood density

GAM and random forest models were used to predict wood density based on the environmental conditions and latitude and longitude of 1 km² grid-cells in tropical South America. Predictions were masked to areas indicated as forest in GLC 2000[75]. Environmental variables for each grid-cell were obtained as described above (e.g. for MCWD, we extracted monthly precipitation and evapotranspiration in each 1km grid-cell, and then calculated the cumulative water balance from the wettest month as described above), except for rugosity which was obtained at 1 km resolution from GTOPO30[76].

We evaluated the spatial applicability of model predictions in three ways (Fig. S7). Firstly, we took 1000 samples (without replacement) of half the dataset, refitted models, and predicted wood density for the entire dataset. We could therefore calculate the standard deviation for each observation in our data. We then related this standard deviation to each explanatory variable using locally-weighted polynomial regression. Taking a standard deviation of 0.05 as an arbitrary threshold for indicating high uncertainty, this allowed us to map areas where predictions were less well constrained (i.e. explanatory variables had values where they were sensitive to subsampling). Secondly, we identified areas with explanatory variables outside the range seen in our training data (i.e. where models are extrapolating to novel absolute conditions). Thirdly, we calculated a multivariate dissimilarity index (DI) following[45]. This method firstly calculates the Euclidean distance between pairs of locations based on their explanatory variables (which have first been scaled for comparability); we did not weight variables by their importance as we wanted the metric to be applicable across different models. The DI is then calculated as the minimum distance to an observation in the training data, divided by the mean distance between training data points. Values of more than one thus indicate points that are more dissimilar to the nearest training data point than the average dissimilarity amongst training data points. We followed Meyer and Pebsma's[45] method for defining binary threshold to denote the area of applicability (i.e. the zone where model validation metrics are expected to give a true measure of performance), noting that the threshold definition is somewhat arbitrary. This approach calculates the DI between each data point and the nearest data point that is not in the same cross-validation fold, and then uses 1.5 times the interquartile range as the upper threshold DI value. This was calculated for both the spatial and non-spatial cross validation approaches.

### Reporting summary

Further information on research design is available in the Nature Portfolio Reporting Summary linked to this article.

## Data availability

The wood density data generated in the study and used to build models of spatial variation in wood density are deposited in https://doi.org/10.5521/forestplots.net/2024_4. Predictions of wood density along with measures of uncertainty and areas of applicability are

deposited in https://doi.org/10.6084/m9.figshare.27118437. Data sources for climate, soil and topography variables used in analyses are listed in Table 1, and extracted values for each plot deposited in https://doi.org/10.5521/forestplots.net/2024_4. Forest cover data are from the Global Land Cover 2000 database [82, https://forobs.jrc.ec.europa.eu/glc2000]. Source data are provided with this paper.

## Code availability

The analysis code is available at https://doi.org/10.5521/forestplots.net/2024_4.

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

## Acknowledgements

This paper is a product of the multiple forest plot networks and their combined records collected over decades and curated at ForestPlots.net. As well as investigators and field leaders included here, we gratefully acknowledge the efforts of several hundred additional botanists, technicians and field assistants who contributed to installation, measurement and identification of trees across South American forests. RAINFOR, PPBio and ForestPlots.net have been supported by numerous people and grants since their inception. For their important contributions to developing the RAINFOR network and its antecedents we are also indebted to Jon Lloyd and Manuel Gloor as well as our late, esteemed colleagues Elisbán Armas, Terry Erwin, Thomas Lovejoy, Alwyn Gentry, Sandra Patiño, Antonio Peña Cruz and Jean-Pierre Veillon. For supporting the networks, we thank the European Research Council (ERC Advanced Grant 291585 – 'T-FORCES'), the Gordon and Betty Moore Foundation (#1656 'RAINFOR', and 'MonANPeru'), the European Union's Fifth, Sixth and Seventh Framework Programme (EVK2-CT-1999-00023 – 'CARBONSINK-LBA', 283080 – 'GEOCARBON', 282664 – 'AMAZALERT'), the Natural Environment Research Council (NE/D005590/1 – 'TROBIT', NE/F005806/1 – 'AMAZONICA', E/M0022021/1 - 'PPFOR'), several NERC Urgency and New Investigators Grants, the NERC/State of São Paulo Research Foundation (FAPESP) consortium grants 'BIO-RED' (NE/N012542/1), 'ECOFOR' (NE/K016431/1, 2012/51872-5, 2012/51509-8), 'ARBOLES' (NE/S011811/1, FAPESP 2018/15001-6), 'SEOSAW' (NE/P008755/1), 'SECO' (NE/T01279X/1), Brazilian National Research Council (PELD/CNPq 403710/2012-0), the Royal Society (University Research Fellowships and Global challenges Awards) (ICA/R1/180100 - 'FORAMA'), the National Geographic Society, US National Science Foundation (DEB 1754647) and Colombia's Colciencias. We thank the National Council for Science and Technology Development of Brazil (CNPq) for support to the Cerrado/Amazonia Transition Long-Term Ecology Project (PELD/441244/2016-5), the PPBio Phytogeography of

Amazonia/Cerrado Transition Project (CNPq/PPBio/457602/2012-0), PELD-RAS (CNPq, Process 441659/2016-0), RESFLORA (Process 420254/2018-8), Synergize (Process 442354/2019-3), the Empresa Brasileira de Pesquisa Agropecuária – Embrapa (SEG: 02.08.06.005.00), the Fundação de Amparo à Pesquisa do Estado de São Paulo – FAPESP (2012/51509-8 and 2012/51872-5), the Goiás Research Foundation (FAPEG/PELD: 2017/10267000329) the EcoSpace Project (CNPq 459941/2014-3), PELD 441572/2020-0, the FATE project (03/12595-7) and several PVE and Productivity Grants. We also thank the "Investissement d'Avenir" program (CEBA, ref. ANR-10LABX-25-01), the São Paulo Research Foundation (FAPESP 03/12595-7, 2016/21043-8) and the Sustainable Landscapes Brazil Project (through Brazilian Agricultural Research Corporation (EMBRAPA), the US Forest Service, USAID, and the US Department of State) for supporting plot inventories in the Atlantic Forest sites in Sao Paulo, Brazil. We thank to the National Council for Technological and Scientific Development (CNPq) for the financial support of the PELD project (441244/2016-5, 441572/2020-0) and FAPEMAT (0346321/2021). We thank Reserva Particular do Patrimônio Natural Serra das Almas for supporting our research at the reserve. This paper also includes plots where recensuses and data assimilation were funded by SECO (NE/T01279X/1), and plots established by Darwin Initiative funded project 20-021, NERC-Newton-FAPESP project Nordeste (NE/N01247X/1; NE/N012550/1) and the USAID funded Partnerships for Enhanced Engagement in Research project. This manuscript is an output of ForestPlots.net Research Project 75A. "Mapping LATAM Forest Wood Density", which is part of the NERC-FAPESP funded project ARBOLES (NE/S011811/1). ForestPlots.net is a meta-network and cyberinitiative developed at the University of Leeds that unites permanent plot records and supports tropical forest scientists. We acknowledge the contributions of the ForestPlots.net Collaboration and Data Request Committee (B.S.M., E.N.H.C., O.L.P., T.R.B., B. Sonké, C. Ewango, J. Muledi, S.L.L., L. Qie) for facilitating this project and associated data management. The development of ForestPlots.net and curation of data has been funded by several grants including NE/B503384/1, NE/N012542/1 - 'BIO-RED', ERC Advanced Grant 291585 - 'T-FORCES', NE/F005806/1 - 'AMAZONICA', NE/N004655/1 - 'TREMOR', NERC New Investigators Awards, the Gordon and Betty Moore Foundation ('RAINFOR', 'MonANPeru'), ERC Starter Grant 758873 -'TreeMort', EU Framework 6, a Royal Society University Research Fellowship, and a Leverhulme Trust Research Fellowship. For supporting M.S. we thank NERC (NE/N012542/1 and NE/W003872/1) and the Royal Society (a Royal Society Global Challenges grant "Sensitivity of Tropical Forest Ecosystem Services to Climate Changes"). F.E. was supported by BJT-FAPESPA Program (Process No. 2021/658588) and the Serrapilheira Institute fellowship/FAPESPA (grant number – R-2401–46863), T.F.D. received support from the Brazilian National Council for Scientific and Technological Development (CNPq - 312589/2022-0 Research Productivity Grant) and FAPESP grant 2015/50488-5, G.W.F was supported by APQ 00031-19 FAPEMIG/Renova and J.P. was supported by a CNPq productivity scholarship (312571/2021-6).

## Author contributions

M.J.P.S., O.L.P. and D.G. designed research, M.J.P.S., O.L.P., D.G., E.A., E.A.O., J.A., E.A.D., L.F.A., A.A., L.A., A.A.-M., E.A., L.A., O.A.M.C., F.B., T.B., O.B., C.B., J.B. Jo.B., E.B., L.B., C.B., D.B., F.B. K.M.B., R.J.W.B., I.S.B., B.B., G.C., J.L.C., D.C., M.A.C., W.C., H.C.dL., L.C., S.C.R., S.C., P.R., V.C.M., J.C., F.C., J.A.C., G.C.V., F.l.C., I.C., Ld.C., M.Bd.M., Jd.A.P., G.D., K.D., M.D., M.Md.E.S., T.F.D., A.D., A.Du., C.D.R., F.E., M.M.E.S., A.E.M., W.F.-R., S.F., T.F., G.W.F., J.F., Y.R.F.N., J.C.G.F., K.G.C., R.G., L.H., R.H., E.N.H.C., W.H.H., M.I., C.A.J., M.K., T.K., J.K., B.K., S.G.L., W.F.L., A.L., S.L.L., M.L.D., G.L.G., W.M., Y.M., L.M.A.M., A.G.M., J.L.M.P., B.S.M., B.H.M.J., J.A.M.V., S.M.R., T.M., W.M., A.M.M., P.M., P.S.M., P.M., S.C.M., M.N., D.N., A.D.L., P.N.V., W.L.O., W.P., N.C.P.C., A.P.G., G.P.M., K.Pd.A., M.P.C., P.J.F.P.R., T.P., G.C.P., J.P., N.C.A.P., M.P., A.P.L., L.P., N.C.Ca.S., H.R.-A., M.R.M., C.R.R., G.R.-T., P.M.S.R., Ds.J.R., T.R.dS., J.R.R.P., G.M.R.M., K.R., Kru., C.R., N.S.R., R.S., R.M.S., T.S., A.S., R.S.B., J.S., M.S.dJ., J.S., M.S., R.C.S., C.S.V., J.O.S., M.S., M.F.S., Y.C.S.-S., P.S., R.M.S.S., T.S., J.T., Ht.S., J.T., R.T., M.T., A.T.-L., W.T., Pvd.H., Md.D.M.V., S.A.V., E.V., J.M.V.C., D.M.Vi., L.J.V., V.A.V., V.W., F.Y.I., P.A.Z., J.A.Z. perfomed research by contributing plot data. M.J.P.S., O.L.P., A.L. and G.C.P. contributed analysis tools. M.J.P.S. analysed data. M.J.P.S. and O.L.P. wrote the first draft, all authors reviewed and edited the paper.

## Competing interests

The authors declare no competing interests.

## Additional information

**Martin J. P. Sullivan** [1,2] ✉, **Oliver L. Phillips** [2], **David Galbraith** [2], **Everton Almeida** [3], **Edmar Almeida de Oliveira** [4], **Jarcilene Almeida**[5], **Esteban Álvarez Dávila**[6], **Luciana F. Alves** [7], **Ana Andrade** [8], **Luiz Aragão**[9], **Alejandro Araujo-Murakami**[10], **Eric Arets** [11], **Luzmila Arroyo**[12], **Omar Aurelio Melo Cruz**[13], **Fabrício Baccaro** [14], **Timothy R. Baker** [2], **Olaf Banki** [15], **Christopher Baraloto** [16], **Jos Barlow** [17], **Jorcely Barroso**[18], **Erika Berenguer** [17,19], **Lilian Blanc** [20,21], **Cecilia Blundo** [22], **Damien Bonal**[23], **Frans Bongers** [24], **Kauane Maiara Bordin**[25], **Roel J. W. Brienen** [2],

Igor S. Broggio[26,27], Benoit Burban[28], George Cabral[5], José Luís Camargo[29], Domingos Cardoso[30,31], Maria Antonia Carniello[32], Wendeson Castro[33], Haroldo Cavalcante de Lima[30], Larissa Cavalheiro[34,35], Sabina Cerruto Ribeiro[36], Sonia Cesarina Palacios Ramos[37], Victor Chama Moscoso[38], Jerôme Chave[39], Fernanda Coelho[2,40], James A. Comiskey[41,42], Fernando Cornejo Valverde[43], Flávia Costa[44], Italo Antônio Cotta Coutinho[45], Antonio Carlos Lola da Costa[46], Marcelo Brilhante de Medeiros[47], Jhon del Aguila Pasquel[48,49], Géraldine Derroire[50], Kyle G. Dexter[51,52,53], Mat Disney[54], Mário M. do Espírito Santo[55], Tomas F. Domingues[56], Aurélie Dourdain[50], Alvaro Duque[57], Cristabel Durán Rangel[58], Fernando Elias[59,60,61], Adriane Esquivel-Muelbert[62], William Farfan-Rios[63], Sophie Fauset[64], Ted Feldpausch[65], G. Wilson Fernandes[66], Joice Ferreira[67], Yule Roberta Ferreira Nunes[68], João Carlos Gomes Figueiredo[55], Karina Garcia Cabreara[69], Roy Gonzalez[70], Lionel Hernández[71], Rafael Herrera[72], Eurídice N. Honorio Coronado[73], Walter Huaraca Huasco[19], Mariana Iguatemy[74], Carlos A. Joly[75], Michelle Kalamandeen[2], Timothy Killeen[76], Joice Klipel[77], Bente Klitgaard[78], Susan G. Laurance[79,80], William F. Laurance[79,80], Aurora Levesley[2], Simon L. Lewis[2,54], Maurício Lima Dan[81], Gabriela Lopez-Gonzalez[2], William Magnusson[82], Yadvinder Malhi[19], Lucio Malizia[83], Augustina Malizia[22], Angelo Gilberto Manzatto[84,85], Jose Luis Marcelo Peña[86], Beatriz S. Marimon[87], Ben Hur Marimon Junior[87], Johanna Andrea Martínez-Villa[88], Simone Matias Reis[36,87], Thiago Metzker[89], William Milliken[90], Abel Monteagudo-Mendoza[91], Peter Moonlight[92,93], Paulo S. Morandi[4], Pamela Moser[94], Sandra C. Müller[77], Marcelo Nascimento[95], Daniel Negreiros[66], Adriano Nogueira Lima[44], Percy Núñez Vargas[96], Washington L. Oliveira[94], Walter Palacios[97], Nadir C. Pallqui Camacho[2,38], Alexander Parada Gutierrez[10], Guido Pardo Molina[98], Karla Maria Pedra de Abreu[99], Marielos Peña-Claros[24], Pablo José Francisco Pena Rodrigues[100], R. Toby Pennington[93,101], Georgia C. Pickavance[2], John Pipoly[102,103], Nigel C. A. Pitman[104], Maureen Playfair[105], Aline Pontes-Lopes[9], Lourens Poorter[24], Nayane Cristina Candida dos Santos Prestes[4], Hirma Ramírez-Angulo[106], Maxime Réjou-Méchain[107], Carlos Reynel Rodriguez[108], Gonzalo Rivas-Torres[109], Priscyla M. S. Rodrigues[110], Domingos de Jesus Rodrigues[34], Thaiane Rodrigues de Sousa[44], José Roberto Rodrigues Pinto[111], Gina M. Rodriguez M.[112], Katherine Roucoux[73], Kalle Ruokolainen[113], Casey M. Ryan[114], Norma Salinas Revilla[115], Rafael Salomão[116,117], Rubens M. Santos[118], Tiina Sarkinen[119], Andressa Scabin[120], Rodrigo Scarton Bergamin[121], Juliana Schietti[44], Milton Serpa de Meira Junior[111], Julio Serrano[122], Miles Silman[69], Richarlly C. Silva[123], Camila V. J. Silva[17,40,124], Jhonathan Oliveria Silva[110], Marcos Silveira[125], Marcelo F. Simon[47], Yahn Carlos Soto-Shareva[91], Priscila Souza[126], Rodolfo Souza[127,128], Tereza Sposito[129], Joey Talbot[130], Hans ter Steege[15,131], John Terborgh[132], Raquel Thomas[133], Marisol Toledo[134], Armando Torres-Lezama[106], William Trujillo[135], Peter van der Hout[136], Maria das Dores Magalhães Veloso[137], Simone A. Vieira[138], Emilio Vilanova[139], Jeanneth M. Villalobos Cayo[140,141], Dora M. Villela[142], Laura Jessica Viscarra[10], Vincent A. Vos[143], Verginia Wortel[144], Francoise Yoko Ishida[80,145], Pieter A. Zuidema[24] & Joeri A. Zwerts[146,147]

[1]Department of Natural Sciences, Manchester Metropolitan University, Manchester, UK. [2]School of Geography, University of Leeds, Leeds, UK. [3]Instituto de Biodiversidade e Floresta, Universidade Federal do Oeste do Pará, Santarém, Brazil. [4]Faculdade de Ciências Agrárias, Biológicas e Sociais Aplicadas, Universidade do Estado de Mato Grosso, Nova Xavantina-MT, Brazil. [5]Departamento de Botânica-CCB, Universidade Federal de Pernambuco, Pernambuco, Brazil. [6]Escuela de Ciencias Agrícolas, Pecuarias y del Medio Ambiente, National Open University and Distance, Bogotá, Colombia. [7]Institute of the Environment and Sustainability, University of California, Los Angeles, Los Angeles, USA. [8]Projeto Dinâmica Biológica de Fragmentos Florestais, Instituto Nacional de Pesquisas da Amazônia, São José dos Campos, Brazil. [9]Divisão de Observação da Terra e Geoinformática (DIOTG), Instituto Nacional de Pesquisas Espaciais (INPE), São José dos Campos, Brazil. [10]Museo de Historia Natural Noel Kempff Mercado, Universidad Autónoma Gabriel René Moreno, Santa Cruz, Bolivia. [11]Vegetation, Forest and Landscape Ecology, Wageningen Environmental Research, Wageningen, The Netherlands. [12]Dirección de la Carrera de Biología, Universidad Autónoma Gabriel René Moreno, Santa Cruz, Bolivia. [13]Universidad del Tolma, Tolima, Colombia. [14]UFAM- Universidade Federal do Amazonas, Manaus, Brazil. [15]Naturalis Biodiversity Center, Leiden, The Netherlands. [16]International Center for Tropical Botany, Department of Biological Sciences, Florida International University, Miami, FL, USA. [17]Lancaster Environment Centre, Lancaster University, Lancaster, UK. [18]Centro Multidisciplinar, Universidade Federal do Acre, Rio Branco, AC, Brazil. [19]Environmental Change Institute, School of Geography and the Environment, University of Oxford, Oxford, UK. [20]Unit Research Forests & Societies, CIRAD, Montpellier, France. [21]Unit Research Forests & Societies, Univ Montpellier, Montpellier, France. [22]Instituto de Ecología Regional, CONICET, Universidad Nacional de Tucumán, Tucumán, Argentina. [23]Université de Lorraine, AgroParisTech, INRAE, UMR Silva, Nancy, France. [24]Forest Ecology and Forest Management Group, Wageningen University, Wageningen, The Netherlands. [25]Ecology Department, Universidade Federal do Rio Grande do Sul, Porto Alegre, Brazil. [26]Laboratório de Ciências Ambientais, Universidade Estadual do Norte Fluminense Darcy Ribeiro (UENF), Campos dos Goytacazes, Brazil. [27]Tropical Ecosystems and Environmental Sciences lab (TREES), Instituto Nacional de Pesquisas Espaciais (INPE), São José dos Campos, Brazil. [28]Ecologie des Forêts de Guyane (ECOFOG), INRA, Kourou, French Guiana. [29]Projeto Dinâmica Biológica de Fragmentos Florestais, Instituto Nacional de Pesquisas da Amazônia, Manaus, Brazil. [30]Jardim Botânico do Rio de Janeiro, Rio de Janeiro, Brazil. [31]Instituto de Biologia, Universidade Federal da Bahia, Salvador, Brazil. [32]Universidade do Estado de Mato Grosso, Nova Xavantina-MT, Brazil. [33]Laboratório de Botânica e Ecologia Vegetal, Universidade Federal do Acre, Rio Branco, AC, Brazil. [34]Núcleo de Estudos da Biodiversidade da Amazônia Mato-grossense, Universidade Federal de Mato Grosso, Sinop, MT, Brazil. [35]Instituto de Ciências Naturais, Humanas e Sociais, Universidade Federal de Mato Grosso, Sinop, MT, Brazil. [36]Centro de Ciências Biológicas e da Natureza, Universidade Federal do Acre, Rio Branco, AC, Brazil. [37]Herbario Forestal, Universidad Nacional Agraria La Molina, Lima, Peru. [38]Jardín Botanico de Misssouri - Perú, Universidad Nacional de San Antonio Abad del Cusco, Cusco, Peru. [39]Laboratoire Evolution et Diversite Biologique, Université Toulouse III - Paul Sabatier, Toulouse, France. [40]BeZero Carbon, London, UK. [41]Inventory & Monitoring Program, National Park Service,

Fredericksburg, VA, USA. [42]Smithsonian Institution, Washington, DC, USA. [43]Proyecto Castaña, Made de Dios, Peru. [44]Instituto Nacional de Pesquisas da Amazônia (INPA), Manaus, Brazil. [45]Universidade Federal do Ceará, Pós-Graduação em Sistemática, Uso e Conservação da Biodiversidade, Fortaleza, Brazil. [46]Instituto de Geociências, Faculdade de Meteorologia, Universidade Federal do Para, Belém, PA, Brazil. [47]Embrapa Genetic Resources & Biotechnology, Brazilian Agricultural Research Corporation (EMBRAPA), Brasília, Brazil. [48]Instituto de Investigaciones de la Amazonia Peruana, Iquitos, Peru. [49]Universidad Nacional de la Amazonia Peruana (UNAP), Iquitos, Peru. [50]UMR EcoFoG (AgroParistech, CNRS, INRAE, Université des Antilles, Université de la Guyane), CIRAD, Kourou, French Guiana. [51]School of GeoSciences, The University of Edinburgh, Edinburgh, UK. [52]Dipartimento di Scienze della Vita e Biologia dei Sistemi, Università di Torino, Turin, Italy. [53]Tropical Diversity Section, Royal Botanic Garden Edinburgh, Edinburgh, United Kingdom. [54]Department of Geography, University College London, London, UK. [55]Departamento de Biologia Geral, Universidade Estadual de Montes Claros, Montes Claros, Brazil. [56]FFCLRP, Universidade de São Paulo, São Paulo, Brazil. [57]Universidad Nacional de Colombia, Medellin, Colombia. [58]Chair of Silviculture, University of Freiburg, Freiburg, Germany. [59]Institute of Biological Sciences, Universidade Federal do Pará, Belém, PA, Brazil. [60]Rede Amazônia Sustentável, Santarém, Brazil. [61]Universidade Federal Rural da Amazônia, Capitão Poço, Pará, Brazil. [62]School of Geography, Earth and Environmental Sciences, University of Birmingham, Birmingham, UK. [63]Department of Biology and Sabin Center for Environment and Sustainability, Wake Forest University, Winston-Salem, North Carolina, United States of America. [64]School of Geography, Earth and Environmental Sciences, University of Plymouth, Plymouth, UK. [65]School of Geography, College of Life and Environmental Sciences, University of Exeter, Exeter, UK. [66]Department of Genetics, Ecology and Evolution, Universidade Federal de Minas Gerais, Belo Horizonte, MG, Brazil. [67]Embrapa Amazônia Oriental, Brazilian Agricultural Research Corporation (EMBRAPA), Brasília, Brazil. [68]Laboratório de Ecologia Vegetal, Universidade Estadual de Montes Claros, Montes Claros, Brazil. [69]Center for Energy, Environment, and Sustainability, Wake Forest University, Winston-Salem, USA. [70]Institutio Alexander von Humboldt, Soledad, Colombia. [71]Universidad Nacional Experimental de Guayana, Bolívar, Venezuela. [72]Instituto Venezolano de Investigaciones Científicas (IVIC), Caracas, Venezuela. [73]School of Geography and Sustainable Development, University of St Andrews, St Andrews, UK. [74]International Institute for Sustainability, Rio de Janeiro, Brazil. [75]Department of Plant Biology, University of Campinas, Campinas, Brazil. [76]Agteca, Santa Cruz, Bolivia. [77]Departamento de Ecologia, Universidade Federal do Rio Grande do Sul, Porto Alegre, RS, Brazil. [78]Department of Accelerated Taxonomy, Royal Botanic Gardens Kew, Richmond, London, UK. [79]College of Science and Engineering, James Cook University, Cairns, Australia. [80]Centre for Tropical Environmental and Sustainability Science, James Cook University, Cairns, Australia. [81]Centro de Pesquisa, Desenvolvimento e Inovação Sul, Instituto Capixaba de Pesquisa, Assistência Técnica e Extensão Rural, Incaper, Cachoeiro de Itapemirim, ES, Brazil. [82]Coordenação da Biodiversidade, Instituto Nacional de Pesquisas da Amazônia (INPA), Manaus, Brazil. [83]Facultad de Ciencias Agrarias, Universidad Nacional de Jujuy, San Salvador de Jujuy, Argentina. [84]Laboratório de Biogeoquímica Ambiental Wolfgang C. Pfeiffer, Universidade Federal de Rondônia, Porto Velho, RO, Brazil. [85]Departamento de Biologia, Universidade Federal de Rondônia, Proto Velho, Brazil. [86]Universidad Nacional de Jaén, Cajamarca, Peru. [87]Programa de Pós-graduação em Ecologia e Conservação, Universidade do Estado de Mato Grosso, Nova Xavantina, MT, Brazil. [88]Université du Québec, Montreal, Canada. [89]Department of Natural Sciences, IBAM - Instituto Bem Ambiental / Grupo Myr, Belo Horizonte, Brazil. [90]Enhanced Partnerships Department, Royal Botanic Gardens Kew, Richmond, London, UK. [91]Jardín Botánico de Missouri, Oxapampa, Peru. [92]Botany, School of Natural Science, Trinity College Dublin, Dublin, Ireland. [93]Royal Botanic Gardens Edinburgh, Edinburgh, UK. [94]Department of Ecology, University of Brasília, Brasília, Brazil. [95]Laboratório de Ciências Ambientais, Universidade Estadual do Norte Fluminense, Campos dos Goytacazes, Brazil. [96]Universidad Nacional de San Antonio Abad del Cusco, Cusco, Peru. [97]Carrera de Ingeniería Forestal, Universidad Tecnica del Norte, Ibarra, Ecuador. [98]Facultad de Ciencias Forestales, Universidad Autónoma del Beni José Ballivián, Riberalta, Bolivia. [99]Instituto Federal de Educação, Ciência e Tecnologia do Espírito Santo, Vitória, Brazil. [100]Diretoria de Pesquisas, Instituto de Pesquisas Jardim Botânico do Rio de Janeiro, Rio de Janeiro, Brazil. [101]College of Life and Environmental Sciences, University of Exeter, Exeter, UK. [102]Environmental Planning and Management, Broward County Parks and Recreation Division, Oakland Park, FL, USA. [103]FL Atlantic University, Blo Sciences, Boca Raton, FL, USA. [104]Keller Science Action Center, Field Museum of Natural History, Chicago, IL, USA. [105]Centre for Agricultural Research in Suriname (CELOS), Paramaribo, Suriname. [106]Universidad de los Andes, Mérida, Venezuela. [107]UMR AMAP, Univ. Montpellier, IRD, CNRS, CIRAD, INRAE, Montpellier, France. [108]Facultad de Ciencias Forestales, Universidad Nacional Agraria La Molina, Lima, Peru. [109]Universidad San Francisco de Quito, Quito, Ecuador. [110]Colegiado de Ecologia, Universidade Federal do Vale do São Francisco, Senhor do Bonfim, Brazil. [111]Departamento de Engenharia Floresta, Universidade de Brasília, Brasília, Brazil. [112]Fundación Ecosistemas Secos de Colombia, Puerto Colombia, Colombia. [113]Department of Biology, University of Turku, Turku, Finland. [114]The University of Edinburgh, Edinburgh, UK. [115]Pontificia Universidad Catolica del Peru, San Miguel, Peru. [116]Universidade Federal Rural da Amazônia/CAPES, Belém, PA, Brazil. [117]Museu Paraense Emílio Goeldi, Belém, PA, Brazil. [118]Departamento de Ciências Florestais, Universidade Federal de Lavras, Lavras, Brazil. [119]Royal Botanic Garden Edinburgh, Edinburgh, UK. [120]Instituto Juruá, Manaus, Brazil. [121]Departamento de Botânica, Universidade Federal do Rio Grande do Sul, Porto Alegre, Brazil. [122]Instituto de Investigación para el Desarrollo Forestal, Universidad de los Andes, Mérida, Venezuela. [123]Campus Avançado Baixada do Sol, Instituto Federal de Educação, Ciência e Tecnologia do Acre, Rio Branco, AC, Brazil. [124]Instituto Nacional de Pesquisa Ambiental da Amazônia (IPAM), Brasília, Brazil. [125]Laboratório de Botânica e Ecologia Vegetal, Centro de Ciências Biológicas e da Natureza, Universidade Federal do Acre, Rio Branco, AC, Brazil. [126]Climate Policy Initiative, Rio de Janeiro, Brazil. [127]Environmental Modeling Program, Texas A&M Transportation Institute, Bryan, TX, USA. [128]Biological and Agricultural Engineering, Texas A&M University, College Station, TX, USA. [129]Department of Natural Sciences, IBAM - Instituto Bem Ambiental, Belo Horizonte, Brazil. [130]Institute for Transport Studies, University of Leeds, Leeds, UK. [131]Quantitative Biodiversity Dynamics, Department of Biology, Utrecht University, Utrecht, The Netherlands. [132]Center for Tropical Conservation, Nicholas School of the Environment, Duke University, Durham, NC, USA. [133]Iwokrama International Centre for Rainforest Conservation and Development, Georgetown, Guyana. [134]Universidad Autónoma Gabriel René Moreno, Santa Cruz de la Sierra, Bolivia. [135]Fundación Reserva Natural La Palmita, Bogotá, Colombia. [136]Van der Hoult Forestry Consulting, Rotterdam, The Netherlands. [137]Departamento de Biologia Geral, Universidade Estadual de Montes Claros, Minas Gerais, Brazil. [138]Núcleo de Estudos e Pesquisas Ambientais, Universidade Estadual de Campinas, Campinas, Brazil. [139]Forests and Climate change Program, Wildlife Conservation Society (WCS), New York, USA. [140]Herbario del Sur de Bolivia, Universidad de San Francisco Xavier de Chuquisaca, Sucre, Bolivia. [141]Escuela de Ciencias Forestales, Universidad Mayor de San simón, Cochabamba, Bolivia. [142]Laboratório de Ciências Ambientais, Universidade Estadual do Norte Fluminense, Campos dos Goytacazes, RJ, Brazil. [143]Instituto de Investigaciones Forestales de la Amazonía, Universidad Autónoma del Beni José Ballivián, Riberalta, Bolivia. [144]Department of Forest Management, Centre for Agricultural Research in Suriname (CELOS), Paramaribo, Suriname. [145]College of Marine and Environmental Sciences, James Cook University, Carins, Australia. [146]Ecology and Biodiversity, Utrecht University, Utrecht, The Netherlands. [147]Forestry, Centre for Agricultural Research in Suriname (CELOS), Paramaribo, Suriname. ✉e-mail: martin.sullivan@mmu.ac.uk

