## [Transparent Peer Review file · Nature Communications]

Variation in wood density across South American tropical forests

Corresponding Author: Dr Martin Sullivan

Version 0:

Reviewer comments:

Reviewer #1

(Remarks to the Author)

Comments on "Variation in wood density across South American tropical forests"

Wood density is an important ecological trait mediating species responses to the environment. Wood density is a fundamental determinant of tree biomass and lack of wood density information can lead to marked discrepancies between remote sensed and ground-based estimates of aboveground biomass. Based on extensive collective effort to measure and identify trees in forests across South America, this paper describe spatial variation in community wood density, and use relationships with environmental variables to map estimated wood density at 1km resolution across tropical and subtropical South American forests. The analyses (1) improves the mapping of wood density at fine-scale variation (1km), (2) refines previously described east-west Amazon gradients in wood density across Amazonia, (3) extends prediction in to Andean, dry and Atlantic forest biomes. These provide important new products for the remote sensing community. I have a few major concerns about the methods and results:

1. The forest inventory plots cover tropical and subtropical South American forests. However, although the East-central Amazon network represents forest area in the Amazon basins, there still limited coverage of plots data for the whole Amazon regions.
2. Fi. 1(b): It would be good to show the number of plots in each region (network). Meanwhile, I would suggest adding the plot information, i.e. location, number of species, number of stems, etc, in the supplemental materials.
3. Lines 249-250: The plots were selected based on being in mature, structurally intact and closed canopy forests (i.e. excluding secondary forests, forests with a known history of logging or burning, and savannah formations). The main objective of this study is to map spatial variation of wood density across South American forests. Since the dataset is mainly mature forests, the uncertainty of the predictions for those secondary forests (savannah formations) areas should be estimated, or at least should be discussed.
4. Lines 268-276: This paper calculated stand-level wood density based on species composition and global wood density dataset (based on Family/genus/species). Three methods were explored, names (1) WD1: the mean value across all taxa in a stand; (2) WD2: the abundance weighted mean of taxa present in the stand; (3) WD3: the basal-area weighted mean wood density. More information about these three methods should be provided in the Methods.
5. Methods and Results, Lines 82-85: This paper built models of generalized additive models (GAMs) and random forests, with three sets of explanatory variables, (1) latitude and longitude only, (2) environmental (climate, soil and topography) variables only, and (3) both environmental and spatial variables. Fig. 2 showed the relationships between wood density and environmental variables. From Fig. 2, the environmental and spatial variable showed weak relationships with wood density. Firstly, the authors did not show the robustness of the models (GAMs, random forests). Secondly, it seems that many relationships are driven by fewer plot data standing on the left or right side, while majority of data points were rather scattered. Thirdly, although the GAMs provide flexibility when fitting the data, it's difficult to provide explanations on the driving mechanisms, for example, why wood density could varies with longitude and latitude with such non-linear ways.
6. Fig. 2: When building the GAMs and random forests models, are the pooled dataset of all plots used? Did the authors try models within different regions?

Reviewer #2

(Remarks to the Author)

The authors present an interesting analyses of wood density estimates in South American forests. I was asked to focus on the methodology in specific. Which I think is fine, but also relatively basic. I think the manuscript has room for improvement, the authors should provide more evidence for the validity of the model(s) by including i) an assessment of extrapolation/spatial validity, ii) include the proper accuracy metrics and assess spatial autocorrelation, and iii) explore the environmental drivers in more detail.

1. The spatial coverage of the dataset is impressive, yet an assessment of its applicability to the spatial extent of the predictions is missing. The authors could use a methodology like the AOA (<https://doi.org/10.1111/2041-210X.13650>), or use an alternative approach to show how well the environmental space captured by the training data describes that of South American forests, and where it doesn't.

2. Lines 105, 111, 350: Rather than the correlation coefficients to evaluate the quality of fit of the model, coefficient of determination values should be reported: $1 - (RSS/TSS)$. Also the fit of the final model ensemble should be reported (i.e. that of the map presented in Fig3), in addition to those of the members of the ensemble.

3. The authors report mean squared errors (MSE), but with the same units as the response variable. This is not possible- units are squared for MSE. It's probably better to use RMSE instead, which is in the same units as the response variable.

4. The authors should include an assessment of spatial autocorrelation. Is there any residual spatial autocorrelation? Including spatial predictors in some of the ensemble members probably helped, but this should be proven. E.g. perform a Moran's I test or plot semivariograms for the residuals.

5. Line 106: figure 5 is absent. I assume this is a scatter plot with predicted vs. observed points with the 1:1 line and a fitted line?

6. Figure 2. I don't think these graphs help much in explaining the story. Rather than the 1:1 relationships of the variables, it might be interesting to include feature importance metrics from e.g. a SHAP analysis that shows how each of the predictors drive the response variable in multivariate space.

7. Abstract: I would maybe rephrase lines 7-8 a bit so it doesn't imply that the dataset comprises in-situ measurements of wood density, when they are rather a product of cross-referencing plot inventory data with species, genus, family or plot level mean values from a published dataset. Maybe omit 'ground-sourced'?

8. How was maximum cumulative water deficit included as a predictor to generate the final maps? Was the per-pixel minimum MCWD value taken?

Version 1:

Reviewer comments:

Reviewer #2

(Remarks to the Author)

I think the authors have addressed the comments (both mine and those from the other reviewer) from the first round of review adequately.

Perhaps the only suggestion would be to integrate the AOA result onto the main map, with regions outside the AO hatched. This would make it much easier for the reader to understand and which areas have the highest confidence.

Dear Editor and Reviewers,

Thank you for providing constructive comments on our manuscript, "Variation in wood density across South American tropical forests". We are pleased that the reviewers considered our work to be "an interesting analysis" which provides "important new products for the remote sensing community".

In the revised manuscript, we have:

- Conducted detailed analyses of the spatial applicability of model predictions (see Reviewer 1 comment 1 and Reviewer 2 comment 1 – our analyses are described in detail in response to the latter comment). These support the applicability of our predictions over most of the area, although identify some important areas (e.g. high-elevation montane forests) where predictions will be more uncertain.
- Added an additional metric of model performance to complement the existing ones (see Reviewer 2 comments 2-3) and assessed residual spatial autocorrelation (see Reviewer 2 comment 4).
- Added new analysis to assess the importance of explanatory variables in multivariate space (see Reviewer 2 comment 6) and assessed the sensitivity of modelled relationships to subsampling data (see Reviewer 1 comment 5).

We have provided a point-by-point response to the reviewers comments below.

REVIEWER COMMENTS

Reviewer #1 (Remarks to the Author):

Comments on "Variation in wood density across South American tropical forests"

Wood density is an important ecological trait mediating species responses to the environment. Wood density is a fundamental determinant of tree biomass and lack of wood density information can lead to marked discrepancies between remote sensed and ground-based estimates of aboveground biomass. Based on extensive collective effort to measure and identify trees in forests across South America, this paper describe spatial variation in community wood density, and use relationships with environmental variables to map estimated wood density at 1km resolution across tropical and subtropical South American forests. The analyses (1) improves the mapping of wood density at fine-scale variation (1km), (2) refines previously described east-west Amazon gradients in wood density across Amazonia, (3) extends prediction in to Andean, dry and Atlantic forest biomes. These provide important new products for the remote sensing community.

I have a few major concerns about the methods and results:

1. The forest inventory plots cover tropical and subtropical South American forests. However, although the East-central Amazon network represents forest area in the Amazon basins, there still limited coverage of plots data for the whole Amazon regions.

RESPONSE: We have now added detailed assessment of how the environmental conditions sampled in our dataset relate to the wider area we are making predictions about. Please see our response to R2 comment 1 for full details.

2. Fi. 1(b): It would be good to show the number of plots in each region (network). Meanwhile, I would suggest adding the plot information, i.e. location, number of species, number of stems, etc, in the supplemental materials.

RESPONSE: We were intending on supplying this with the input data as part of the accompanying data package, but can provide this as a SI table if desired (our reluctance for having this in the SI is that this would involve duplicating information from the data package).

3. Lines 249-250: The plots were selected based on being in mature, structurally intact and closed canopy forests (i.e. excluding secondary forests, forests with a known history of logging or burning, and savannah formations). The main objective of this study is to map spatial variation of wood density across South American forests. Since the dataset is mainly mature forests, the uncertainty of the predictions for those secondary forests (savannah formations) areas should be estimated, or at least should be discussed.

RESPONSE: Thank you, this is an important point. This analysis is focused on old-growth forests, so expanding to secondary forest would be a major undertaking involving different networks, plots and authors. We have added some discussion on this.

Lines 236-241: *"It is important to note that our predictions are for mature, closed canopy forests, so should not be used for secondary forests. Wood density is expected to be lower in secondary forests [2], although in some seasonally dry and montane forests wood density can decline with succession [44, 55]. These differences in trajectories of wood density between forest types may be explained by differing successional mechanisms [44], so we may expect large-scale spatial patterns in secondary forests to differ from the old-growth forest patterns described here."*

4. Lines 268-276: This paper calculated stand-level wood density based on species composition and global wood density dataset (based on Family/genus/species). Three methods were explored, names (1) WD1: the mean value across all taxa in a stand; (2) WD2: the abundance weighted mean of taxa present in the stand; (3) WD3: the basal-area weighted mean wood density. More information about these three methods should be provided in the Methods.

RESPONSE: Yes. We have now expanded our description of these methods to provide more information about them. See lines 323-339.

5. Methods and Results, Lines 82-85: This paper built models of generalized additive models (GAMs) and random forests, with three sets of explanatory variables, (1) latitude and longitude only, (2) environmental (climate, soil and topography) variables only, and (3) both environmental and spatial variables. Fig. 2 showed the relationships between wood density and environmental variables. From Fig. 2, the environmental and spatial variable showed weak relationships with wood density. Firstly, the authors did not show the robustness of the models (GAMs, random forests). Secondly, it seems that many relationships are driven by fewer plot data standing on the left or right side, while majority of data points were rather scattered. Thirdly, although the GAMs provide flexibility when fitting the data, it's difficult to provide explanations on the driving mechanisms, for example, why wood density could varies with longitude and latitude with such non-linear ways.

RESPONSE: Thank you. We now assess the sensitivity of the relationships in the GAM and random forest models to changing the input data. We take 1000 samples of half the data, and refit the GAM and random forest models to each subset. We then use these models to make predictions to the full dataset, and assess the standard deviation of predictions for each plot location. This is especially useful for assessing the potential for relationships to be driven by a few data points. We assess how the standard deviation of model predictions relate to each explanatory variable. While uncertainty in relationships does increase for some variables (e.g. at high lightning frequencies), the random forest models in particular were generally robust to subsampling. We have indicated areas in the spatial extent of predictions where uncertainty revealed by this subsampling is high (Fig S7, see response to R2 comment 1).

This analysis is presented in Fig. S6 and described in Lines 176-177 and 180-187.

Latitude and longitude can relate to wood density as (1) dispersal limitation could lead to spatially structured but non-environmentally driven patterns in species distributions and (2) it can capture spatially structured effects of unknown environmental variables that influence wood density. It is likely that these spatial patterns (especially those through the second mechanism) are quite complex and variable, so potentially could lead to non-linear patterns. We had discussed how latitude and longitude affect wood density in lines 203-217. The complexity of GAM relationships with other variables was constrained by limiting the maximum degrees of freedom (see lines 393-397).

6. Fig. 2: When building the GAMs and random forests models, are the pooled dataset of all plots used? Did the authors try models within different regions?

RESPONSE: Yes, the pooled dataset of all plots was used. Models were also constructed for each region and predictions of these were closely related to those using the pooled dataset (Fig. S5).

Reviewer #2 (Remarks to the Author):

The authors present an interesting analyses of wood density estimates in South American forests. I was asked to focus on the methodology in specific. Which I think is fine, but also relatively basic. I think the manuscript has room for improvement, the authors should provide more evidence for the validity of the model(s) by including i) an assessment of extrapolation/spatial validity, ii) include the

proper accuracy metrics and assess spatial autocorrelation, and iii) explore the environmental drivers in more detail.

1. The spatial coverage of the dataset is impressive, yet an assessment of its applicability to the spatial extent of the predictions is missing. The authors could use a methodology like the AOA (<https://doi.org/10.1111/2041-210X.13650>), or use an alternative approach to show how well the environmental space captured by the training data describes that of South American forests, and where it doesn't.

RESPONSE: Thank you for these suggestions. We now provide several assessments of the applicability of our models to the spatial extent of predictions:

1. We repeatedly refitted models to random subsets of the data (1000 samples of half the plots) and assessed how the standard deviation of predictions related to each explanatory variable (see response to R1 comment 5). This enabled us to identify areas where model predictions were less well constrained.
2. We assessed which areas had explanatory variable values outside the range seen in the training data.
3. We used Meyer and Pebesma's dissimilarity index (DI) to assess the area of applicability (AOA) of our models, using thresholds derived from both the spatial and non-spatial cross-validation methods.

Most of these show that our models are applicable to the vast majority of the spatial extent of predictions (although the environmental conditions of very high elevation Andean forests are not well captured). The most restrictive assessment of applicability comes from the AOA approach using a DI threshold based on the DIs seen in the non-spatial cross-validation, which indicates that some dry forests and parts of lowland Amazonia are also outside the AOA. We suspect that the outlier removal method used to select the DI threshold is rather harsh, as many areas with DI values represented in the cross-validation folds subsets are considered outside the AOA. We also found no clear pattern between prediction error and DI (Fig. R1), so while we are happy to present the AOA, we are not convinced the threshold choice is well supported for our data.

Figure R1. Relationship between prediction error and DI. The red line shows a LOWESS fit between absolute prediction error and the dissimilarity index.

These assessments are presented in Figure S7, and described in Line 179-190.

2. Lines 105, 111, 350: Rather than the correlation coefficients to evaluate the quality of fit of the model, coefficient of determination values should be reported: $1-(RSS/TSS)$. Also the fit of the final model ensemble should be reported (i.e. that of the map presented in Fig3), in addition to those of the members of the ensemble.

RESPONSE: We have added the coefficient of determination (calculated as $1-(RSS/TSS)$) and the fit of the final model ensemble to these comparisons. Because we are now presenting more information, we have switched from presenting this as a figure to presenting it as a table (i.e. Fig. 4 becomes Table 2).

We do still present correlation metrics as they provide important additional information about model performance by showing whether observed values increase with predicted values. We also note that the caret R package, which is widely used for this sort of model evaluation, bases their calculation of the coefficient of determination on the square of the correlation coefficient, so it is a widely used evaluation metric.

3. The authors report mean squared errors (MSE), but with the same units as the response variable. This is not possible- units are squared for MSE. It's probably better to use RMSE instead, which is in the same units as the response variable.

RESPONSE: Sorry for the typographical mistake. The values reported are now RMSE.

4. The authors should include an assessment of spatial autocorrelation. Is there any residual spatial autocorrelation? Including spatial predictors in some of the ensemble members probably helped, but this should be proven. E.g. perform a Moran's I test or plot semivariograms for the residuals.

RESPONSE: We assessed residual spatial autocorrelation using semivariograms and present them as Fig. S9. There is no clear pattern of residual spatial autocorrelation, with an increase in semivariance only evident at very large distances which presumably are capturing biome boundaries.

5. Line 106: figure 5 is absent. I assume this is a scatter plot with predicted vs. observed points with the 1:1 line and a fitted line?

RESPONSE: We apologise, this was meant to be a reference to Figure 4. (The scatter plot of predicted vs observed points is presented in Fig. S5, but was not meant to be referred to from this line.)

6. Figure 2. I don't think these graphs help much in explaining the story. Rather than the 1:1 relationships of the variables, it might be interesting to include feature importance metrics from e.g. a SHAP analysis that shows how each of the predictors drive the response variable in multivariate space.

RESPONSE: We now show feature importance from SHAP analysis instead of this figure. We have moved the original Figure 2 into the supporting materials as we do think it is important to retain these plots of the raw data somewhere, to improve transparency of what is going on behind the models.

7. Abstract: I would maybe rephrase lines 7-8 a bit so it doesn't imply that the dataset comprises in-situ measurements of wood density, when they are rather a product of cross-referencing plot inventory data with species, genus, family or plot level mean values from a published dataset. Maybe omit 'ground-sourced'?

RESPONSE: We have changed this as suggested (omitting ground-sourced) to avoid any incorrect implications.

8. How was maximum cumulative water deficit included as a predictor to generate the final maps? Was the per-pixel minimum MCWD value taken?

RESPONSE: It was calculated in the same way as described for its derivation as an explanatory variable, using monthly precipitation and evapotranspiration extracted for each pixel. We have expanded the description of this in the methods (Lines 437-440).

Response to reviewers: Variation in wood density across South American tropical forests

Dear Editor and Reviewers,

Thank you for your further evaluation of our manuscript. The only comments received on the last version were from reviewer two, below.

Reviewer 2: *I think the authors have addressed the comments (both mine and those from the other reviewer) from the first round of review adequately.*

Perhaps the only suggestion would be to integrate the AOA result onto the main map, with regions outside the AO hatched. This would make it much easier for the reader to understand and which areas have the highest confidence.

Response: We agree this is a good idea. The fine resolution of the map means that hatching effectively obscures the data underneath, so rather than add hatching we have now changed Figure 3 to show areas outside the area of occurrence in grey. The original Figure 3 without the AOA overlay has been moved to the supporting materials.

We have additionally reviewed the manuscript to add clarifications of sample sizes and definitions of boxplots as requested in the author checklist, and to divide the results section into subsections. On reviewing the data and code prior to deposition we noticed that an additional regional split had been used in the spatial cross-validation and regional fitting analyses. We have re-run these analyses to ensure consistency of regional definitions with those presented in Fig. 1 – this has resulted in minor changes to Table 2 and Fig. S5, but does not change any result in a substantive way. We have also added a small statement relating to inclusion and ethics in global research. While we would be happy to have this as a separate section, we felt this fitted in nicely with the existing text on research networks so have instead integrated it with existing text in lines 405-410.

We believe other queries have been addressed in the author checklist, but please do let us know if there is anything outstanding.

Sincerely,

Martin Sullivan and co-authors.